# Associations of Implicit and Explicit Sexual Double Standard Endorsement and Sexual Assertiveness with Sexual and Interactional Competence in Emerging Adults

**DOI:** 10.3390/bs14090790

**Published:** 2024-09-09

**Authors:** Andrea Grauvogl, Ron Pat-El, Jacques J. D. M. van Lankveld

**Affiliations:** Faculty of Psychology, Open University of the Netherlands, P.O. Box 2960, 6401 DL Heerlen, The Netherlands; ron.pat-el@ou.nl (R.P.-E.); jacques.vanlankveld@ou.nl (J.J.D.M.v.L.)

**Keywords:** explicit sexual assertiveness, implicit sexual assertiveness, sexual double standard endorsement, sexual and interactional competence

## Abstract

In this study, among emerging adults, we investigated the interrelationships of explicit and implicit measures of sexual assertiveness (SA) and sexual double standard endorsement (SDS) on the one hand, and different aspects of sexual and interactional competence (SAIC) on the other hand, using Partial Least Squares Path Modeling (PLS-PM) of cross-sectional data. Participants were 159 sexually active, heterosexual individuals in the Netherlands between 18 and 25 years. No exclusion criteria were used. The Sexual Competence and Interaction Competence in Youth and lifetime number of sexual partners were used to measure SAIC. Explicit SA was measured using the Hurlbert Index of Sexual Assertiveness, while Explicit SDS was assessed using the Scale for the Assessment of Sexual Standards in Youth. Two implicit association tests were performed to measure implicit SA and SDS. Participants accessed these computerized reaction time tasks via a secure online data collection platform. Results showed a strong association between the latent factors of sexual attitudes and SAIC. Greater SA and lower SDS were associated with a greater competence level. No gender effects were found.

## 1. Introduction

In this study, among emerging adults, we investigated the interrelationships of explicit and implicit measures of sexual assertiveness (SA) and sexual double standard endorsement on the one hand, and different aspects of sexual and interactional competence (SAIC) on the other hand. The terms sexual competence and interactional competence refer to different but related constructs in the field of sexuality. Recent conceptualizations of sexual competence focus on various aspects of sexual behavior [1,2,3,4]. Sexual competence consists of cognitive factors and skills aimed at solving problems that arise in different sexual contexts, including communication ability, refusal and resistance, conflict resolution, and interpersonal negotiation skills [5,6]. While these authors conceive of sexual competence as consisting of personal characteristics and skills that act causally to achieve sexual health, others view sexual competence more as the outcome state of such characteristics and skills. For instance, Palmer et al. [3] describe sexual competence at first intercourse as encompassing the aspects of protection by contraception, consensuality (willingness of both partners), autonomy (intercourse was not due to external influences such as alcohol or peer pressure), and acceptability of timing as perceived by both individuals. In line with the WHO definition of sexual health [7], some definitions of sexual competence also include the cognitive, emotional and interactional skills to achieve the experience of sexual pleasure and satisfaction [2,8,9]. These recent conceptualizations are now widely accepted among researchers, although they do not necessarily coincide with lay definitions of sexual competence in youth [10,11].

Among researchers, both psychological and interactional skills are seen as relevant to sexual competence [1,4]. Psychological skills refer to the ability to identify one’s own sexual needs and limits. Interactional skills include the ability to communicate one’s needs and limits and to adequately assess and appraise the partner’s responses, including the partner’s consent. Gold and colleagues [4] emphasized the importance of interactional competence as an intermediary to realizing one’s sexual competence in sexual interactions. Interactional competence includes communicative and social skills, and mental and behavioral strategies that serve to arrange sexual encounters with a partner in a mutually satisfactory manner [1]. Although the descriptions of sexual competence and interactional competence clearly show considerable overlap, interactional competence refers more to action aspects in an interaction context, including sexual communication, while sexual competence points to individual psychological factors and protective outcomes of sexual interactions.

SAIC are thus conceptualized as consisting of a wide range of cognitive and behavioral skills. In the present study, we selected four constituent factors to represent SAIC: sexual communication, pleasurable sexuality, contraceptive use, and limited lifetime number of sexual partners. Three of these factors are measured using the Sexual Competence and Interaction Competence in Youth (SCICY) [12], and represent sexual communication skills, positive feelings about sex, and behavioral competence with condom use. A high lifetime number of sexual intercourse partners is seen as entailing increased risk of sexually transmitted infections, including HIV, and as a threat to sexual health in adolescents and emerging adults [13,14]. The number of lifetime sexual partners is measured with a single question.

For the purpose of developing interventions aimed at enhancing young people’s SAIC, it is important to increase our knowledge of (modifiable) factors related to SAIC [15]. Some determinants of SAIC have previously been investigated. Knowledge of one’s sexual preferences [6,16], but not general sexual knowledge, and social and behavioral competence [6] were found to predict sexual health in emerging adults. Relationship factors, including the partner’s autonomy-supportive versus controlling behavior, were also found to affect sexual competence [17]. The predictors of sexual health outcomes in these studies showed only modest effect sizes, explaining up to 33% of the variance in the full prediction model [16], leaving room for additional explanatory factors. In addition to sexual knowledge and relational factors, a wide array of other determinants has been suggested, including among others: school-based education on sexuality and relationships [11], general level of education [18], sexual literacy [2], sexual assertiveness [9,19,20] and sexual double standard endorsement [21]. In this study we focused on the latter two factors as potential predictors of SAIC.

### 1.1. Sexual Double Standard Endorsement

The term “sexual double standard” (SDS) is used to describe a set of normative expectations for engaging in romantic and sexual behavior, which differ for young men and women [22]. A central aspect of the SDS is gender differences in sexual assertiveness (SA); young men are expected to be sexually active, dominant, and take sexual initiative, whereas young women are expected to be sexually reserved, submissive, and passive [20,23]. Endorsement of the SDS is associated with various negative sexual health outcomes [21]. In young men, SDS was found to be associated with greater acceptance of rape myths [24], beliefs that dating violence is acceptable, and sexually violent behavior [25,26,27]. For both young men and women, it is related to early sexual debut [28,29] and higher STI/HIV infection risk [30,31]. The impact of SDS seems particularly negative for girls [21]. The sexual passivity implicated in SDS predicts more negative and fewer positive emotions related to sex in young women [20], as well as lower sexual satisfaction [32,33] and more sexual problems. In a Spanish study, men but not women who favored egalitarian gender roles in sexuality reported greater sexual satisfaction [34]. In a study among mostly African American adolescents, girls had more gender equitable attitudes [23]. Participants with more equitable gender norms reported perpetrating significantly less relationship abuse. Gender equitable attitudes were also significantly correlated with condom negotiation self-efficacy.

Over time, the SDS concept has evolved; some researchers argue that the SDS no longer exists [35,36], or only under specific conditions [37]. Other researchers observed a reverse SDS [38,39,40,41], although traditional sexual script adherence was recently found to still impact behavior [42]. Based on previous research outcomes that imply multiple associations of SDS with sexual health outcomes, we considered it an important candidate for inclusion in the current study.

### 1.2. Sexual Assertiveness

Assertiveness in sexual encounters is considered an important correlate of SAIC [9,19,20]. Greater SA was associated with greater sexual satisfaction among female undergraduate students in the US [43], although romantic and casual sexual relationships in youth did not differ in the actors’ level of SA in a Dutch study [9]. The hypothesized interrelatedness of SA and SDS was tested in a cross-sectional study among Ecuadorian emerging adults aged 18 to 30 [18]. SA was positively associated with overall mental well-being and relationship satisfaction in women but not in men. In turn, SA was negatively predicted by SDS, especially among less educated women and men. In a study in Germany, Klein et al. [42] conceptualized SDS as acceptance vs. rejection of traditional gender scripts and performed an online vignette study in which adult male participants were instructed to imagine themselves single in a bar situation with a woman. The scenario described the woman as displaying either assertive, timid, or controlling sexual behavior. Confirming the traditional gender script and SDS, men were found to perceive women who behaved sexually assertively as nonconformist. Their judgment of the assertive woman was less positive than that of women who expressed sexual timidity or control. The authors concluded from the findings of a second study using the same design and methods among both women and men that it was SA, not sexual script deviation, that determined their evaluations of the imagined target [42]. SA, therefore, seems a strong candidate for inclusion in the present study of correlates of SAIC in young adults. To our knowledge, no study has yet examined the associations between both SDS and SA and SAIC simultaneously.

### 1.3. Explicit Versus Implicit Factors

In addition to deliberate attitudes and intentions, attitudes that operate at an automatic cognitive level may be relevant in SAIC. Automatic cognitions, assessed with the help of indirect measures, have been found to predict behavior in various fields of human functioning, including sexual functioning (e.g., [44]), condom (non-)use [45,46], aggressive behavior [47], alcohol use (e.g., [48]), physical activity [49], and psychopathology (e.g., [50]). There is broad consensus that measuring implicit associations is able to bypass introspective access and socially desirable biases [51]. In several reviews and meta-analyses, implicit associations were found to add limited but significant proportions of explained variance to directly measured deliberate attitudes [49,52]. Thus, implicit measurements may provide different and additional [51], although not necessarily more (or less) accurate, information on SA and SDS. Good theoretical understanding of the determinants of SAIC, and sufficient empirical support of this theory, are building blocks for future efficacious interventions [53]. In the present study, we aim to answer the following research question: What are the interrelationships between SAIC on the one hand, and SDS and SA on the other hand? We also explored the bivariate associations between the key variables, and whether the associations differed in young women vs. men.

## 2. Method

This study is part of a larger research project. Among others, previous studies in this project investigated the psychometric aspects of two implicit association tests for measuring implicit sexual assertiveness and implicit sexual double standard endorsement, which were required for conducting the present study (see the Instruments section below).

### 2.1. Sample

Undergraduate psychology students at the Open Universiteit recruited participants from their personal circles of acquaintances. Sexually active, heterosexual, emerging adults between the ages of 18 and 25 with a good command of the Dutch language were eligible for participation. No exclusion criteria were used.

### 2.2. Instruments

*Demographics.* Participants reported their age (years), gender (female, male, other), and sexual orientation (on a five-point scale ranging from ‘1 = exclusively attracted to men’ to ‘5 = exclusively attracted to women’).

*Sexual and Interactional Competence*. The latent construct of SAIC was comprised four factors: sexual communication skills, emotional well-being regarding sexuality, and behavioral competence with condom use, all measured using the Sexual Competence and Interaction Competence in Youth (SCICY) [12], as well as the lifetime number of sexual partners. The SCICY is a 26-item self-report questionnaire aimed at measuring sexual and interaction competence in adolescents. It is organized into eight subscales, three of which were selected for the present purpose: sexual communication skills (6 items), positive feelings about sex (5 items), and behavioral competence with condom use (2 items). Responses are given using 5-point Likert scales. The answer options differ per item group. For items in the selected subscales that were formulated as statements, answer options were: 1 “I always agree”, 2 “I mostly agree”, 3 “I sometimes agree/disagree”, 4 “I mostly disagree”, and 5 “I always disagree”. For items asking participants to report behavioral initiatives, the answer options were: 1 “I would always dare to do that”, 2 “I would usually dare to do that”, 3 “I would/wouldn’t dare to do that sometimes”, 4 “I wouldn’t dare to do that most of the time”, and 5 “I would never dare to do that”. For self-descriptive items, answer options were 1 “Never applies to me”, 2 “Usually does not apply to me”, 3 “Sometimes applies/does not apply to me”, 4 “Usually applies to me”, 5 “Always applies to me”. Higher scores represent higher levels of communication skills, positive feelings about sex, and competence with contraceptive use. The subscales were found to have adequate internal consistency (alphas ranging from 0.77 to 0.84), and moderate to excellent test–retest reliability (communication about sex: r = 0.76, *p* < 0.001; positive feelings about sex: r = 0.54, *p* < 0.001; contraceptive use: r = 1.00, *p* < 0.001) [12].

*Lifetime Number of Sexual Partners* was reported by participants as their response to a single question: “How many people have you had sex with (approximately) in your life? By sex, we mean everything from touching or caressing genitals to sexual penetration”. Responses were recorded as integers. Participants were allowed to refrain from answering the question without providing a reason.

*Implicit Sexual Double Standard Endorsement.* An IAT [54] was designed to measure Implicit SDS (SDS-IAT) [41]. Target category labels were ‘male’ versus ‘female’, and attribute category labels were ‘sexually active’ versus ‘sexually passive’. Stimulus words were presented in the middle of the screen. Words representing the target categories were Dutch male (i.e., Bram, Tim, Rob, Jan) and female names (i.e., Emma, Lieke, Julia, Roos). Words representing the ‘sexually active’ attribute category were ‘sexual’, ‘exciting’, ‘experienced’, and ‘daring’ (in Dutch: ‘seksueel’, ‘opwindend’, ‘ervaren’, ‘uitdagend’). Words representing the ‘sexually passive’ attribute category were ‘biding’, ‘reserved’, ‘cautious’, and ‘modest’ (in Dutch: ‘afwachtend’, ‘terughoudend’, ‘voorzichtig’, ‘bescheiden’). The SDS-IAT was found to possess adequate psychometric properties [41]. Attenuation-corrected alphas (range: 0.65–0.70) demonstrated acceptable internal consistency. A multitrait-multimethod analysis of the SDS-IAT demonstrated low convergent validity with Explicit SDS, supporting theoretical expectations. Divergent validity was confirmed by the absence of significant correlations with conceptually unrelated concepts, except for extraversion in female participants.

*Implicit Sexual Assertiveness.* An IAT was designed to measure Implicit SA (SA-IAT) [55]. Attribute categories were ‘I’ (in Dutch: ‘ik’) versus ‘Other’ (in Dutch: ‘een ander’), and target categories were ‘sexually assertive’ (in Dutch: ‘seksueel assertief’) versus ‘sexually compliant’ (in Dutch: ‘seksueel meegaand’). Stimuli presented in the center of the screen were either words representing the ‘I’ attribute category (‘I’, ‘me’, ‘self’, ‘mine’ [in Dutch: ‘ik, mij, zelf, mijn’]), or the ‘other’ attribute category (you, their, your [in Dutch: ‘je, jij, hun, jouw’]), or words associated with the ‘sexually assertive’ (confident, initiative-taking, leading, adventurous [in Dutch: ‘zelfverzekerd, initiatiefrijk, leidend, avontuurlijk’]) versus ‘sexually compliant’ (following, agreeable, subordinate, dependent [in Dutch: ‘volgend, toegeeflijk, onderdanig, afhankelijk’]) target categories. The SA-IAT was found to possess adequate psychometric properties [55]. Attenuation-corrected alphas (range: 0.61–0.70) demonstrated acceptable internal consistency. A multitrait-multimethod analysis of the SA-IAT demonstrated low convergent validity with Explicit SA, supporting theoretical expectations. Divergent validity was confirmed by the absence of significant correlations of Implicit SA with conceptually unrelated concepts, except for extraversion in the full sample and the female subsample.

The labels of the target and attribute categories in both IATs were permanently visible in the upper-left and upper-right corners of the screen. Correct responses were defined as key presses with which stimuli were placed into the category they were a priori considered to represent. After a correct response, the next stimulus was presented after a 250 ms interval. After an incorrect response, a red X replaced the stimulus and remained on the screen until the correct key was pressed. The IATs were organized into five blocks. To familiarize participants with the procedure, they started with a practice run of 16 trials presenting only stimuli from the target category (e.g., gender in the SDS-IAT: ‘male’ versus ’female’; block 1). Next was a practice block of 16 trials (block 2), in which both target and attribute stimuli were presented, followed by a test block of 48 trials (block 3). In these two blocks, one of two possible combinations of target and attribute categories (e.g., female + sexually passive; male + sexually active) was mapped to the response keys (‘z’ and ‘m’ on a QWERTY keyboard). In the final blocks, a practice block (block 4) and a test block (block 5), including the same number of trials, the reverse combination was presented (e.g., male + sexually passive; female + sexually active). Two versions of each IAT were made. They differed in the order of presentation of blocks 2 + 3 and 4 + 5, allowing for the investigation of potential order effects. Random allocation ensured that half of the participants started with each version. Testing for order effects revealed a within-IAT order effect for the SDS-IAT, but not for the SA-IAT. No between-IAT order effects were found.

*Explicit Sexual Double Standard Endorsement.* Explicit SDS was assessed using the Scale for the Assessment of Sexual Standards in Youth (SASSY) [56]. The 19-item SASSY was designed as a multifaceted measure of SDS in young people and was found to have a single underlying factor. An example item is ‘Sometimes a boy should apply some pressure to a girl to get what he wants sexually’. Answers were given on a 6-point scale ranging from ‘1 = Completely disagree’ to ‘6 = Completely agree’. In previous research, good reliability was found among Dutch adolescents (α = 0.90) [56] and young adults (Study 1: α = 0.88, Study 2: α = 0.89) [56]. A mean score was calculated from all items. Higher scores represent a higher level of SDS. In the present sample a Cronbach’s α of 0.85 indicated good internal consistency.

*Explicit Sexual Assertiveness*. To assess Explicit SA, a selection of eight items from the Hurlbert Index of Sexual Assertiveness (HISA) was used [57,58,59]. Items were selected while taking the factor structure of the scale into account, ensuring that items from both the ‘initiation’ (3 items) and ‘no shyness/refusal’ dimensions (5 items) were included [60]. An example item from the ‘initiation’ dimension is ‘I think I am open with my partner about my sexual needs’. An example item from the ‘no shyness/refusal’ dimension is ‘It is hard for me to say no, even when I do not want sex’. Answers were given on a 5-point scale ranging from ‘0 = Never’ to ’4 = Always’. A mean score was calculated from all items. Higher scores represent a higher level of SA. Satisfactory reliability for both the ‘initiation’ (α = 0.77) and the ‘no shyness/refusal’ (α = 0.72) subscales was found in the present sample.

### 2.3. Procedure

Ethical approval was obtained from the University Ethics Committee (filed under reference FETC-14024). Participant anonymity and data security were ensured using the university’s secure online data acquisition platform. Participants completed the questionnaires and performed the computer tasks in the comfort of their own home using an online research platform to reduce bias due to social demands. Online assessment of IAT has been found to produce robust findings that do not differ from assessments in a lab setting [61,62]. Participants first completed either the SDS-IAT or an IAT assessing Implicit SA. The order of both IATs was randomly assigned. After completing the IATs, participants completed the questionnaires of the study in a fixed order. Completing the study took about 30 min. After finishing the final task, participants received an email with a debriefing message and a digital gift voucher worth EUR 10.

### 2.4. Statistical Analysis

The D600 algorithm was employed to calculate scores for Implicit SDS and Implicit SA. Only test block data were used. Reaction times (RTs) below 400 ms were discarded, and those higher than 2500 ms were replaced with 2500 before calculating the mean RTs. Error trial RTs were replaced with the mean RT of the participant’s correct responses in the same block in which the error occurred, plus a 600 ms penalty. The D600 index score was calculated as the difference between the mean RTs, divided by the pooled standard deviation with the exception of the attribute practice block. Negative scores reflect higher Implicit SDS, while positive scores reflect higher Implicit SA.

Prior to analysis, the data were screened for errors in data entries, missing values, and abnormal distributions and values. In the dataset, missing values were identified in the following variables: SA-IAT (26 missing), SDS-IAT (27 missing), lifetime number of sex partners (21 missing), Explicit SA (8 missing), Explicit SDS (6 missing), sexual communication skills (8 missing), positive feelings about sex (8 missing), and behavioral competence with condom use (8 missing). To mitigate the impact of missing data on the analysis, the mice package [63] in R version 4.2.2 [64] was employed. The predictive mean matching method was used for imputation, generating five multiple imputations to ensure a robust and reliable dataset for subsequent analyses.

To test the relationship between the study’s latent constructs, we utilized Partial Least Squares Path Modeling (PLS-PM), employing the PLS-PM package in R [64]. PLS-PM is a non-parametric statistical technique used for structural equation modeling (SEM) that differs from covariance-style SEM by focusing on the relationships between latent variables and observed indicators [65]. Unlike traditional covariance-based SEM, which emphasizes covariances and variances among variables, PLS-PM is a variance-based approach that prioritizes explaining the variance in the endogenous constructs. This makes PLS-PM particularly well-suited for modeling complex relationships in small sample sizes or situations with high collinearity. Additionally, PLS-PM is advantageous for formative measurement models, where latent constructs are considered to be caused by their indicators rather than causing them. This flexibility in handling formative constructs distinguishes PLS-PM, allowing researchers to effectively model and understand situations where variables are more causally driven by their observed indicators.

The analysis encompassed two latent variables with a formative measurement model. The dataset, consisting of 159 cases and 8 manifest variables, was standardized. The PLSPM was executed with a centroid weighting scheme and a maximum of 100 iterations. The two latent variables, sexual attitudes and SAIC, were defined as exogenous and endogenous blocks, respectively. Unidimensionality assessments revealed satisfactory internal consistency reliability for both blocks, with composite reliability (α) and eigenvalues supporting the robustness of the latent constructs.

## 3. Results

### 3.1. Descriptive Statistics and Preliminary Analyses

Data were collected from a convenience sample of 159 participants (N_women_ = 98; N_men_ = 55; N_missing_ = 6) living in the Netherlands or Flanders, Belgium. Demographic features, their categorization, and scores on sexual variables are shown in Table 1. The mean ages of female (22.2 ± 1.9 years) and male participants (21.8 ± 1.9 years) were not significantly different. Moderate and high education levels prevailed and were not significantly different between genders. Male participants were significantly more often single or dating than female participants, while the latter were more often in a committed relationship. Response patterns differed between women and men regarding their sexual attraction to the other versus their own gender and their relationship status. A larger proportion of female participants reported feeling ‘mostly’ attracted to the other gender compared to male participants.

Descriptive statistics of the study variables and 95% CIs for differences between young women and men on these variables are presented in Table 2. Young men reported greater sexual communication skills, more positive feelings about sex, and higher explicit sexual assertiveness (SA) but lower Implicit SA. Compared to young women, men exhibited stronger implicit endorsement of the sexual double standard (SDS), whereas women endorsed a reverse Implicit SDS. No significant differences were found in Explicit SDS between genders.

The three factors of the SAIC construct measured using the SCICY, correlated significantly in the full sample (*r*s between 0.292 and 0.411) and in the female subsample (*r*s between 0.299 and 0.486; see Table 3). In the male subsample, only sexual communication skills and behavioral competence with condom use were found to correlate (*r* = 0.292). Lifetime number of sexual partners did not correlate with the other factors. Sexual communication skills, positive feelings about sex, and behavioral competence with condom use correlated positively with Explicit SA in the full sample, while positive feelings about sex correlated negatively with Explicit SDS, indicating that stronger SDS (negative scores) came with stronger positive feelings. This correlation was stronger in young men than in young women. Lifetime number of sexual partners correlated negatively with behavioral competence with condom use in young men, but this association was absent in young women. A negative correlation between Explicit SDS and Explicit SA was found, indicating that greater SDS came with greater SA, with the correlation being significant only in the young women’s subsample.

### 3.2. Hypothesis Testing

We employed Partial Least Squares Path Modeling (PLS-PM) to investigate predictive relationships between sexual attitudes and SAIC.

Two cases with high counts of lifetime sexual experiences (36 and 60, respectively) were identified. The PLS-PM analysis was conducted twice: once including all cases and once excluding these two extreme cases. The difference in results was negligible, leading us to present the analysis including all cases.

### 3.3. The Outer Model

In PLS-PM, the outer model examines the relationships between observed variables (manifest variables) and the underlying latent constructs they represent. These relationships help us understand how well the observed variables contribute to and define the latent constructs, ultimately aiding in the meaningful interpretation of the relationships between the latent constructs.

### 3.4. Sexual Attitudes

The outer model shows how different factors contributed to sexual attitudes Specifically, Explicit SA emerged as a crucial positive indicator of sexual attitudes (weight: 0.916), meaning that higher values of Explicit SA are associated with higher Sexual Attitudes. In contrast, both Explicit SDS and Implicit SDS are weak negative indicators (weight: −0.222 and −0.212, respectively), indicating a slight decrease in sexual attitudes with higher levels of these factors. Implicit Sexual Assertiveness (Implicit SA) has a very weak positive impact (weight: 0.144). Overall, sexual attitudes were primarily indicated by Explicit SA.

### 3.5. Sexual and Interactional Competence

For Sexual and Interactional Competence (SAIC), positive feelings about sex played the most significant role (weight: 1.07). Sexual communication skills (weight: −0.182) and the number of sexual partners (weight: −0.136) had minimal influence. Notably, competence with contraceptive use had almost no impact and was negatively associated (weight: −0.0001). Thus, SAIC was primarily determined by positive feelings about sex.

### 3.6. Cross-Loadings

Cross-loadings examine how well each observed variable (manifest variable) is associated with the latent constructs they are supposed to measure, as well as with other latent constructs. They help verify the specificity and validity of the constructs. In this model, it pertains to how observed variables that were indicators for sexual attitudes also related to latent SAIC, and vice versa.

Cross-loadings showed (in Figure 1) that the observed variables correlated highly with their intended construct and had low correlations with other constructs, confirming their specificity. An exception was Explicit SA, which showed a strong negative association with sexual attitudes (−0.949) and a modest positive association with SAIC (0.559). Similarly, positive feelings about sex were strongly linked with SAIC and had a modest negative association with Sexual Attitudes. Other cross-loadings were generally low (<0.3) and lower than the loadings on the intended constructs.

### 3.7. Inner Model

The inner model, which can be seen as the “regression” part of the PLS-PM, examines how sexual attitudes predict SAIC. There was a strong negative relationship (β = −0.589, t(1) = −9.13, *p* < 0.001), indicating that higher sexual attitudes were linked to lower SAIC.

Combining information from the inner and outer models helps interpret this negative relationship. Given the dominant role of positive feelings about sex in the SAIC construct, with a high weight (1.07), and that sexual attitudes are primarily indicated by explicit SA (weight = 0.916), the negative relationship between SA and SAIC can be largely described as that individuals with higher sexual attitudes, driven by higher explicit sexual assertiveness, tend to have lower SAIC, largely because those more assertive individuals have less positive feelings about sex.

### 3.8. Multigroup Analysis

Table 3 displays correlations between variables, highlighting differences in the strength of these relationships between men and women. To better understand these differences, we conducted a separate analysis for each group (men and women) using our statistical method (PLS-PM).

The results are presented in Figure 2 and Figure 3. Overall, the models for men and women were similar, but there was a stronger negative path coefficient of sexual attitudes on SAIC in young men (β = −0.77) compared to young women (β = −0.56). Figure 4 illustrates these differences, particularly in how sexual communication skills, Implicit SA, Implicit SDS, and the lifetime number of sexual partners contribute to the models.

Here are some key points:-Explicit Sexual Assertiveness (ESA): ESA was a crucial positive indicator for sexual attitudes in both genders. For women, the weight was very high (0.916), and for men, it was even slightly higher (0.931). This shows that ESA played a significant role in shaping sexual attitudes in both men and women.-Positive feelings about sex: This variable had a dominant positive contribution to SAIC for both genders, with a weight of 1.086 for women and 0.931 for men, showing that positive feelings about sex played a significant role in SAIC for both men and women.-Lifetime number of sexual partners: For young men, having more sexual partners had a small positive contribution to SAIC (0.254). For young women, it had a small negative contribution (−0.213).-Sexual communication skills: These skills have a small positive contribution to young women’s SAIC (0.28) but a very small negative contribution for young men (−0.10).-Implicit Sexual Double Standard (ISDS): This variable had a negative contribution to SA for women (−0.385) but a positive contribution for men (0.221).-Implicit Sexual Assertiveness (ISA): ISA showed a positive contribution to SA for women (0.244) but a negative contribution for men (−0.293).-Condom competence: This factor had a minimal positive impact on SAIC for both genders, with a weight of 0.066 for women and 0.125 for men, indicating a relatively minor role in shaping SAIC.

## 4. Discussion

In this study we examined the interrelationships between sexual attitudes, encompassing SDS and SA on the one hand, and SAIC on the other. Sexual attitudes were measured both as deliberate cognitions and as automatic/implicit associations. We also explored the bivariate associations between the key variables.

In support of our expectations, the association between the latent constructs of sexual attitudes and SAIC (the ‘inner model’) was significant and showed a substantial effect size. The use of PLS-PM enabled us to test a formative measurement model, assuming that the latent construct of SAIC is caused by the latent construct of Sexual Attitudes, in line with general models of behavior regulation that assign a causal role to context-specific attitudes [66,67,68].

Positive feelings about sex made the largest contribution to the latent construct of SAIC, followed by behavioral competence with condom use and sexual communication skills, In contrast, the lifetime number of sexual partners made a very limited contribution, differing from findings in earlier research [13,14].

The contributions of the selected variables to the latent construct of sexual attitudes were found to vary considerably. The largest contribution was made by Explicit SA, followed by Explicit SDS, while the contributions of both implicit variables were small. The findings with regard to explicit attitudes aligned with earlier research on the associations of attitudes and SAIC [18,30,31,34,42,43].

The contribution of the lifetime number of sexual partners to the definition of SAIC was found to be low in the present study. The literature on sexual health in adolescents and emerging adults has suggested that a high number of sexual partners during their lifetime poses a threat to sexual health because—obviously—this increases the risk of acquiring sexually transmitted infections, including HIV, as a result of increased exposure to pathogens [13,14]. Indeed, a high number of sexual partners in adolescence was found to be related to increased risk of acquiring STIs and unintended pregnancy, as well as to absent parental monitoring, contact with delinquent peers, and problematic behavior in studies by Van Ryzin, Johnson [14] and Valois, Oeltmann [13]. However, these researchers did not assess other aspects of SAIC, including Positive feelings about sex, allowing for the possibility that in their studies, the lifetime number of sexual partners was also not related to other, experiential and behavioral aspects of SAIC.

The very low contribution of both implicit attitudes contrasted with our expectations. Previous research demonstrated that implicit associations with sexual stimuli predicted several aspects of sexuality, including sexual functioning [44] and condom use [45]. Unfortunately, the present study does not provide empirical evidence to explain these findings. However, it can be speculated that implicit attitudes toward sexual stimuli influence sexual behavior and sexual responses under conditions in which the attentional capacity required to make informed decisions is greatly reduced by ‘the heat of the moment’ [46,69,70,71]. Lower cognitive processing capacity greatly increases the possible influence of implicit associations on these outcomes [70,71], as the more serial processing of deliberate attitudes requires sufficient attention and ample processing capacity, whereas automatic cognizing is capable of processing parallel streams of information [72], although the absence of this effect has also been found [73]. Implicit attitudes regarding SDS and SA might play a role in earlier phases of behavioral preparation where attitudes guide the perception and interpretation of one’s situation [74], and in which cognitive processing capacity is less taxed.

### Strengths and Limitations

The present study used explicit, as well as implicit instruments to measure SDS and SA and included a relatively large sample size. The inclusion of young men in this study is a strength, as the differential effect of sexual attitudes on SAIC in young men, as compared with young women, has, to our knowledge, received scarce research attention (but see [23,75]). Although the employed statistical approach allows testing formative models, in which latent constructs are assumed to be caused by their predictors rather than causing them, the cross-sectional design of the study does not enable strong causal claims. Future studies using a prospective design are warranted to support the indications of a causal path between sexual attitudes and SAIC.

## 5. Conclusions

We conclude that the present cross-sectional study suggests the existence of a causal path between sexual attitudes and sexual and interactional competence in emerging adults. This is observed most prominently as the effect of sexual attitudes on the level of positive feelings about sex, followed by the effects of sexual attitudes on behavioral competence with condom use and on sexual communication skills. The largest contribution to the latent construct of sexual attitudes is made by explicit sexual assertiveness, followed by explicit endorsement of the sexual double standard, while implicit measures of these constructs contributed very little.

## Figures and Tables

**Figure 1 behavsci-14-00790-f001:**
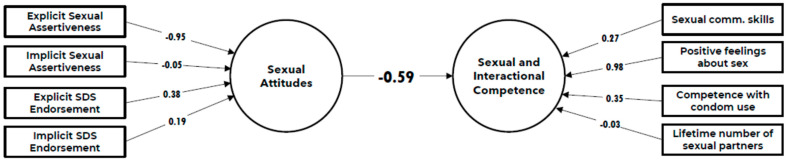
Path model in the full sample.

**Figure 2 behavsci-14-00790-f002:**
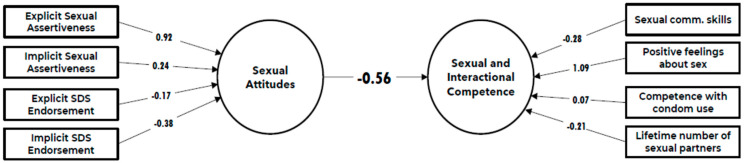
Path model in the female subsample.

**Figure 3 behavsci-14-00790-f003:**
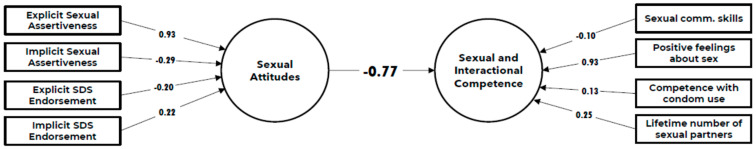
Path model in the male subsample.

**Figure 4 behavsci-14-00790-f004:**
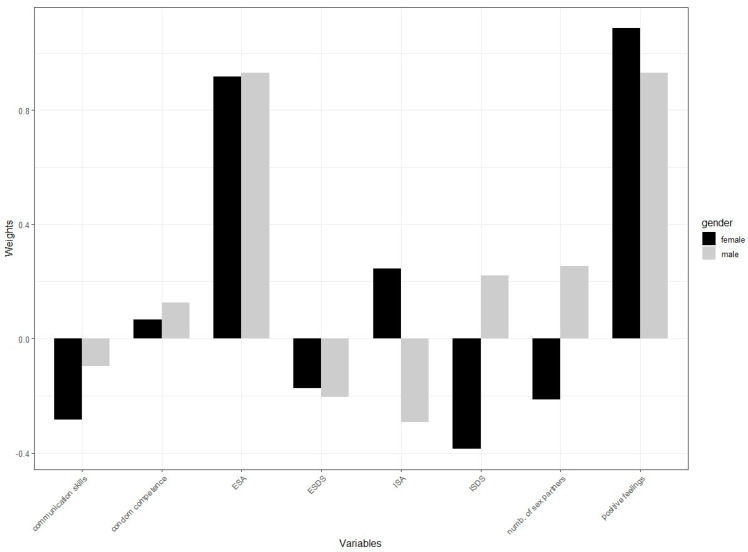
Comparison of path loadings in the female and male subsamples. ESA: Explicit Sexual Assertiveness; ESDS: Explicit Sexual Double Standard endorsement; ISA: Implicit Sexual Assertiveness; ISDS: Implicit Sexual Double Standard endorsement.

**Table 1 behavsci-14-00790-t001:** Demographic characteristics.

	Men (M(SD); %)	Women (M(SD); %)
Age	21.8 (1.9)	22.2 (1.9)
Sexually Attracted To ^1^		
Only Men	9.1	69.8
Mostly Men	1.8	24.0
Men and Women	3.6	4.2
Mostly Women	10.9	0.0
Only Women	72.7	1.0
I Don’t Know (Yet)	0.0	1
I Don’t Want to Disclose	1.8	0.0
Education Level		
Lower	6.3	2.2
Intermediate	70.8	60.7
Higher	22.9	37.1
Relationship Status ^2^		
Single	38.2	29.2
Dating	20.0	11.3
Committed Relationship	40.0	54.3
Married	1.8	2.0

^1^: *χ*^2^(6) = 116.798, *p* < 0.001; ^2^: *χ*^2^(3) = 10.020, *p* < 0.05.

**Table 2 behavsci-14-00790-t002:** Means and standard deviations among young men and women.

	Full Sample (n = 126–153)		Young Men (n = 43–55)		Young Women (n = 82–98)		95% CI Gender Difference	
	M	SD	M	SD	M	SD	Lower	Upper
Sexual communication skills (SCICY) ***	2.80	0.45	3.02	0.33	2.68	0.46	0.21	0.47
Positive feelings about sex (SCICY) *	4.21	0.84	4.36	0.58	4.11	0.95	0.03	0.52
Behavioral competence with condom use (SCICY)	3.85	1.18	3.93	1.14	3.81	1.21	−0.28	0.52
Lifetime number of sexual partners	6.54	3.99	7.12	7.07	6.21	6.97	−1.56	3.37
Implicit SDS Endorsement (IAT-SDS; seconds) ***	−0.01	0.39	−0.16	0.40	0.07	0.35	−0.37	−0.10
Implicit Sexual Assertiveness (SA-IAT; seconds) *	−0.12	0.39	−0.25	0.39	−0.06	0.37	−0.32	−0.04
Explicit SDS Endorsement (SASSY)	2.23	0.57	2.32	0.66	2.18	0.51	−0.07	0.34
Explicit Sexual Assertiveness (HISA) *	2.93	0.66	3.06	0.53	2.85	0.72	0.00	0.41

* *p* < 0.05; *** *p* < 0.001; SCICY: Sexual Competence and Interaction Competence in Youth; SASSY: Scale for the Assessment of Sexual Standards in Youth; HISA: Adapted Hurlbert Index of Sexual Assertiveness.

**Table 3 behavsci-14-00790-t003:** Bivariate correlations in the full sample and the female and male subsamples.

	1	2	3	4	5	6	7	8
Full Sample (N = 115–153)								
1 Sexual communication skills (SCICY)	-							
2 Positive feelings about sex (SCICY)	0.411 ^c^	-						
3 Behavioral competence with condom use (SCICY)	0.292 ^c^	0.341 ^c^	-					
4 Lifetime number of sexual partners	0.106	0.125	−0.095	-				
5 Implicit SDS Endorsement (IAT-SDS; seconds)	−0.023	−0.111	0.105	0.094	-			
6 Implicit Sexual Assertiveness (SA-IAT; seconds)	−0.067	−0.014	0.030	−0.137	0.077	-		
7 Explicit SDS Endorsement (SASSY)	−0.001	−0.199 ^a^	−0.116	−0.042	−0.148	−0.006	-	
8 Explicit Sexual Assertiveness (HISA)	0.161 ^a^	0.539 ^c^	0.204 ^a^	0.007	−0.024	−0.150	−0.207 ^a^	-
Female (Top Right: N = 74–98) and Male (Bottom Left: N = 41–55) Subsamples								
1 Sexual communication skills (SCICY)	-	0.486 ^c^	0.299 ^b^	0.189	0.057	−0.045	0.032	0.197
2 Positive feelings about sex (SCICY)	−0.024	-	0.380 ^c^	0.159	−0.106	0.057	−0.188	0.483 ^c^
3 Behavioral competence with condom use (SCICY)	0.292 ^a^	0.235	-	0.013	0.047	0.046	−0.098	0.251 ^a^
4 Lifetime number of sexual partners	−0.149	−0.013	−0.311 ^a^	-	0.191	−0.071	−0.114	−0.049
5 Implicit SDS Endorsement (IAT-SDS; seconds)	0.161	0.036	0.236	0.010	-	−0.158	−0.047	0.123
6 Implicit Sexual Assertiveness (SA-IAT; seconds)	0.163	−0.066	0.012	−0.242	0.281	-	0.067	−0.169
7 Explicit SDS Endorsement (SASSY)	−0.203	−0.337 ^a^	−0.161	0.050	−0.231	−0.056	-	−0.245 ^a^
8 Explicit Sexual Assertiveness (HISA)	−0.163	0.710 ^c^	0.070	0.105	−0.192	0.028	−0.219	-

^a^ = *p* < 0.05; ^b^ = *p* < 0.01; ^c^ = *p* < 0.001.

## Data Availability

The original data presented in the study are openly available in https://osf.io/upxns/?view_only=f437af4db174453eb798367f5d376904 (accessed on 4 September 2024).

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
