# Peer review of "Associations of Implicit and Explicit Sexual Double Standard Endorsement and Sexual Assertiveness with Sexual and Interactional Competence in Emerging Adults"

_behavsci, 2024, doi:10.3390/bs14090790_

Round 1
Reviewer 1 Report
Comments and Suggestions for Authors
I fear that those who are not strong in statistical techniques will find this article of little value. The inner and outer models became quite confusing. The discussion does little to alleviate this problem. In the end I, as a reader, found myself wondering what was the significance of the article. This article, as written requires the reader to be quite sophisticated statistically and deeply immersed in the specialty topic presented. I fear this will sharply limit the appeal and influence of the article.
Author Response
Thank you for taking the time to read our article and your feedback. We have tried to make the results and discussion section more comprehensible. We have made the following changes:
- We have added a bulletwise summary of the results.
- We added a more detailed description of the demographic data, while referring to Table 1 for how the demographic variables were categorized.
- In the discussion section we have expanded the text, concerning the reasons for the low contribution of implicit measures, to provide more context and hypothetical explanations for these findings.
- A section on the speculative explanation of the low contribution of lifetime number of sexual partners was added to the Discussion.
Reviewer 2 Report
Comments and Suggestions for Authors
The abstract provides a clear overview of the study's objectives, methods, participants, and key findings. It clearly states the aim of the study: to investigate the interrelationships between explicit and implicit measures of sexual assertiveness (SA) and sexual double standard endorsement (SDS) with aspects of sexual and interactional competence (SAIC). It mentions the method used (Partial Least Squares Path Modeling of cross-sectional data) and the participants (159 sexually active, heterosexual individuals aged 18-25).
Suggestions for improvement:
The description of the methodology, while clear, could benefit from more detail on how the implicit association tests were conducted and how the Partial Least Squares Path Modeling was applied. This additional information would provide a better understanding of the study's rigor.
More demographic details about the participants, such as the geographic location or any inclusion/exclusion criteria, would help contextualize the findings.
The introduction is comprehensive and well-structured, providing a thorough background on the study's key concepts and previous research. It thoroughly covers the concepts of sexual competence, interactional competence, sexual assertiveness (SA), and sexual double standard endorsement (SDS).It cites a wide range of relevant literature, providing a strong foundation for the study.
Suggestions for improvement
While the introduction is comprehensive, the flow could be improved by organizing the sections more clearly. For example, separating the discussion of SA and SDS into distinct paragraphs could enhance readability. Use subheadings to break up the text and guide the reader through different sections.
Althpogh the background information is detailed, the connection to the current study's specific aims and hypotheses could be made more explicit earlier in the introduction. This would help readers understand the direct relevance of the background information.
Method and Procedure Commentary
The sample selection of sexually active, heterosexual, emerging adults aged 18-25 with good command of Dutch is clearly defined.
Improvements: Consider including exclusion criteria to refine the sample, such as excluding participants with previous mental health diagnoses that might affect sexual behavior or attitudes.
Procedure - Online data collection is efficient and reduces social desirability bias. The use of random assignment for the IATs helps control for order effects.
Suggestions:
- Consider conducting a pilot study to refine the online platform usability.
- Provide more information on how randomization was implemented and checked.
- Ensure participant anonymity and data security are explicitly addressed.
RESULTS
Strengths:
The descriptive statistics section provides a thorough overview of the demographic features and sexual variables of the sample, allowing for a clear understanding of the participant characteristics and initial patterns in the data. The comparison between men and women in terms of sexual communication skills, feelings about sex, and sexual assertiveness is well-structured and highlights significant gender differences effectively. The use of Partial Least Squares Path Modeling (PLSPM) and multigroup analysis enhances the robustness of the findings by examining predictive relationships and variations between male and female subsamples.
Areas for Improvement:
The section could benefit from a more concise presentation of key findings to enhance readability. Summarizing the most critical points in bullet points or a summary table might help. Including a more detailed explanation of the demographic variables (e.g., education level) and how they were categorized could provide clearer context for the results. Addressing the missing data (Nmissing = 6) and its potential impact on the results would strengthen the validity of the conclusions drawn.
Results Section:
The results section provides a detailed account of the relationships between sexual attitudes, sexual assertiveness, and sexual and interactional competence (SAIC), highlighting significant findings. The application of Partial Least Squares Path Modeling (PLSPM) allows for a sophisticated analysis of the relationships between latent constructs, supporting the validity of the findings. The identification of the contributions of different variables to the latent constructs of SAIC and sexual attitudes is well-done, emphasizing the key factors involved.
Areas for Improvement:
While the distinction between explicit and implicit measures is valuable, the reasons for the low contribution of implicit measures need more exploration. Providing more context or hypotheses for these findings would enhance understanding.
Incorporating a comparative discussion on why certain findings differ from previous research (e.g., lifetime number of sexual partners) could offer deeper insights.
